

# Impact of salinity on element incorporation in two benthic foraminiferal species with contrasting Magnesium contents

Esmee Geerken[1], Lennart Jan de Nooijer[1], Inge van Dijk[1,2] & Gert-Jan Reichart[1,3]

[1]Department of Ocean Systems, NIOZ-Royal Netherlands Institute for Sea Research, and Utrecht University, Den Burg, The Netherlands

[2]Currently at: UMR CNRS 6112 LPG-BIAF, Bio-Indicateurs Actuels et Fossiles, Angers University, France

[3]Faculty of Geosciences, Utrecht University, Utrecht, The Netherlands

*Correspondence to*: Esmee Geerken (esmee.geerken@nioz.nl)

**Abstract.** Accurate reconstructions of seawater salinity could provide valuable constraints for studying past ocean circulation, the hydrological cycle and sea level change. Controlled growth experiments and field studies have shown the potential of foraminiferal Na/Ca as a direct salinity proxy. Incorporation of minor and trace elements in foraminiferal shell carbonate varies, however, greatly between species and hence extrapolating calibrations to other species needs validation by additional (culturing) studies. Salinity is also known to impact other foraminiferal carbonate-based proxies, such as Mg/Ca for temperature and Sr/Ca for seawater carbonate chemistry. Better constraints on the role of salinity on these proxies will improve their reliability. Using a controlled growth experiment spanning a salinity range of 20 units and analysis of single chamber element composition using laser ablation-ICP-MS, we here show that Na/Ca correlates positively with salinity in two benthic foraminiferal species (*Ammonia tepida* and *Amphistegina lessonii*). The Na/Ca values differ between the two species, with an approximately 2-fold higher Na/Ca in *Amphistegina* than in *Ammonia*, which coincides with an offset in their Mg content (~35 mmol/mol versus ~2.5 mmol/mol for *A. lessonii* and *A. tepida*, respectively). Despite the offset in average Na/Ca values, the slopes of the Na/Ca-salinity regressions are similar between these two species. In addition, Mg/Ca and Sr/Ca are positively correlated with salinity in cultured *A. tepida*, but do not show a correlation to salinity for *A. lessonii*. Electron microprobe mapping of incorporated Na and Mg of the cultured specimens shows that within chamber walls of *A. lessonii*, Na/Ca and Mg/Ca occur in elevated bands in close proximity to the primary organic lining. For specimens of *A. tepida*, Mg-banding shows a similar pattern to that in *A. lessonii*, albeit that variation within the chamber wall is much less pronounced. Also Na-banding is much less prominent in this species. The less prominent banding and lower Mg and Sr contents of *A. tepida* are likely related to the absence of an inter-element correlation within experimental conditions.





## 1. Introduction

Seawater salinity varies over time and space as a function of continental ice volume, evaporation, precipitation and river runoff. Reconstructions of salinity could provide important constraints on past ocean circulation, the hydrological cycle and glacial-interglacial sea level changes. Currently, most reconstructions of salinity are indirect and based on the correlation between salinity and $\delta^{18}O_{water}$, assuming this relationship to be constant over space and time. An independent salinity proxy may reduce the uncertainties inherently associated with such approaches and should preferably be based on one of the main components of seawater salinity, for instance sodium (Na). Results from a culture study showed that foraminiferal calcitic Na/Ca (Na/Ca$_{cc}$) correlates positively and linearly with salinity for the low-Mg benthic symbiont-barren species *Ammonia tepida*, with a slope of 0.22 between salinities 30 and 38.6 (Wit et al., 2013). Various culture studies earlier showed that also Mg/Ca is affected by salinity, but respond more strongly to temperature (Lea et al., 1999; Dissard et al., 2010b; Nürnberg et al., 1996; Hönisch et al., 2013). Although an effect of salinity on foraminiferal Sr/Ca$_{cc}$ has been reported in some studies (Kısakürek et al., 2008; Dissard et al., 2010b; Wit et al., 2013) other studies did not find a relation between salinity and foraminiferal Sr/Ca (Dueñas-Bohórquez et al., 2009; Diz et al., 2012; Allen et al., 2016) which is thought to mainly reflect sea water carbonate chemistry (Keul et al., 2017) and temperature (Nürnberg et al., 1996; Lea et al., 1999; Raja et al., 2007). Hence, an independent salinity proxy would not only be useful for constraining past (changes in) salinity, but also improve temperature reconstructions based on Mg/Ca$_{cc}$ and reconstructions of past sea water carbonate chemistry based on Sr/Ca.

Following the culture-based Na/Ca$_{cc}$-salinity calibration for *A. tepida* (Wit et al., 2013), a culture study with planktonic symbiont-bearing species also showed a significant linear relationship for *Globigerinoides ruber (Allen et al., 2016).* Although no significant relationship was observed in this study for *G. sacculifer* (Allen et al., 2016)*, a recent field calibration observed positive linear relationships for both species (Mezger et al., 2016). Still, the Na/Ca-salinity sensitivities observed between the different species and studies differed considerably (ranging from a change in 0.074 to 0.66 mmol/mol in Na/Ca$_{cc}$ for a change in 1 salinity unit). Whereas Wit et al. (2013) suggested an incorporation mechanism similar to that observed in inorganic calcite, field and culture studies also show that different species of foraminifera have varying calcite chemistries, thereby resulting in the need of species-specific calibrations similar to many other foraminiferal trace metal-based proxies (e.g. Elderfield and Ganssen, 2000; Rosenthal et al., 2000; Anand et al., 2003; Bemis et al., 1998; Toyofuku et al., 2011). Mg/Ca$_{cc}$ values for example are different between groups of low-Mg-, high-Mg hyaline and porcelaneous foraminifera (Toyofuku et al., 2000; Segev and Erez, 2006; Raja et al., 2007), which also seems to be reflected in other co-precipitated cations (De Nooijer et al., 2017). Hence, calibration of Na/Ca$_{cc}$ as a function of salinity for other species is not only necessary to test the applicability of this novel proxy for other groups of foraminifera, but also allows testing whether monovalent cations follow the inter-species trends described for divalent cations (Terakado et al., 2010).

Here we calibrated Na-, Mg- and Sr-incorporation in the intermediate-Mg calcite, benthic symbiont-bearing, tropical foraminifer *Amphistegina lessonii* and the low-Mg calcite, symbiont-barren, intertidal species *Ammonia tepida* over a salinity range of 20 units (from 25 to 45) and compare obtained ratios



with existing calibrations. The chemical composition of the calcite was determined by Laser Ablation
Inductively Coupled Plasma Mass Spectrometry (LA-ICP-MS), providing insights in concentrations
and variability in concentrations between specimens and between single chambers. To investigate intra-
specimen variability at the scale of the chamber wall we also performed Electron Probe Micro Analysis
(EPMA), mapping the Ca, Na and Mg distribution throughout the chamber wall for specimens of both
species cultured.
**2. Methods**
**2.1 Collection of foraminifera and culture set-up**
Surface sediment samples containing foraminifera (*A. lessonii*) were collected from the Indo-Pacific
Coral Reef aquarium in Burgers' Zoo (Arnhem, The Netherlands; Ernst et al., 2011) and a tidal flat
near Den Oever, the Wadden Sea (*A. tepida*). Sediment was stored in aerated aquaria at 25°C (*A.*
*lessonii*) and 10°C (*A. tepida*) with a day/night cycle of 12/12 hours, similar to conditions in the coral
reef aquarium and Wadden Sea. From both stocks, living specimens, recognized by chambers that were
filled with yellow cytoplasm and pseudopodial activity, were isolated.
Living specimens were placed in groups of 25 individuals in Petri dishes with approximately 70 ml of
North Atlantic surface seawater (0.2 µm filtered) and fed with fresh cells of the algae *Dunaliella*
*salina*. After reproduction, which occurred in approximately 2/3 of all incubated specimens, 2-3
chambered juveniles were isolated (De Nooijer et al., 2014). The use of specimens from reproduction
events guarantees that virtually all chambers present at the end of the experiment were produced under
the culture conditions (De Nooijer et al., 2014). Strains of specimens of the reproduction events (2-10
individuals) were divided over Petri dishes with approximately 10 ml culture medium and stored in a
temperature controlled incubator set at 25 °C with a day/night cycle of 12/12 hours. The culture media
in the Petri dishes were replaced once every week, after which specimens were fed with approximately
1 ml concentrated and freeze-dried *Dunaliella salina* diluted with the culture medium for each salinity
to avoid changes in salinity. After 6-8 weeks, specimens were harvested and transferred to microvials
to clean the specimens' carbonate shells from cell material. Organic matter was removed by adding
70% $H_2O_2$ buffered with 0.1M $NH_4OH$ at 90 °C and gentle ultrasonication for 1 min. Specimens were
subsequently rinsed 3 times with double deionized water, dried in a laminar flow cabinet, after which
their size was determined (i.e. the maximum diameter crossing the center of the specimen). The
specimens were thereafter stored until geochemical analyses (LA-ICP-MS; 2.2.2 and EPMA; 2.4).
**2.2 Analytical methods**
**2.2.1 Culture media preparation and chemistry**
In total, 50 L of seawater with a salinity of 50 was prepared by sub-boiling 0.2 µm filtered North
Atlantic seawater for 48 hours at 45 ºC. Subsequently, culture media were obtained by diluting this
high-saline seawater with double de-ionized seawater in batches of approximately 10L with salinity



increasing from 25 to 45 in steps of 5 units, resulting in 5 unique salinity conditions. Using a single
batch of concentrated seawater to subsequently dilute to the desired salinities ensures constant element
to Ca ratios. Culture media were stored in Nalgene containers and kept in the dark at 10 °C. Seawater
pH was determined with a pH meter (pH110, VWR). Subsamples were taken prior to and at the end of
the experiment and analyzed for DIC and element concentrations to monitor the effect of sub-boiling
on the seawater's inorganic carbon chemistry and element composition (Table 1). Subsamples for DIC
were collected in headspace-free vials and conserved with a saturated $HgCl_2$ solution (10µl $HgCl_2$/10
ml sample). DIC measurements were performed on an autoanalyzer spectrometric system (TRAACS
800; (Stoll et al., 2001). This analysis requires only a small amount of sample, while yielding high
accuracy (±2 µmol/kg) and precision (±1.5 µmol/kg). The minor and major elemental composition of
the culture media was measured using a sector field ICP-MS (Element2, Thermo Scientific) by
sampling 1 ml from the culture media and dilution by a factor 300 with 0.14 M $HNO_3$ (Table 1).
**Table 1.** Experiment culture media measurements per salinity condition.

| Experiment | Na/Ca$_{sw}$ mol/mol | Mg/Ca$_{sw}$ mol/mol | Sr/Ca$_{sw}$ mmol/mol | Salinity | DIC µmol/kg | pH | [CO$_3^{2-}$] mmol/kgSW | ΩCa |
|---|---|---|---|---|---|---|---|---|
| S25 | 48.84 | 5.61 | 9.37 | 25.2 | 1087.3 | 8.32 | 164.90 | 4.28 |
| S30 | 49.79 | 5.69 | 9.45 | 30.3 | 1305.3 | 8.28 | 205.98 | 5.15 |
| S35 | 48.56 | 5.51 | 9.04 | 35.2 | 1512.0 | 8.22 | 258.84 | 6.22 |
| S40 | 48.50 | 5.62 | 9.19 | 40.0 | 1734.4 | 8.17 | 267.23 | 6.16 |
| S45 | 48.90 | 5.73 | 9.21 | 45.2 | 1947.4 | 8.10 | 284.67 | 6.23 |


### 123   2.2.2 Foraminiferal calcite chemistry

Specimens were fixed on a laser ablation-stub using double sided tape, carefully positioning them to
allow ablation of the last chambers (Appendix A). Element concentrations of individual chambers were
measured with LA-ICP-MS (Reichart et al., 2003). The last 1-3 chambers of each specimen were
ablated using a circular spot with a diameter of 80 μm (NWR193UC, New Wave Research) in a helium
environment in a New Wave TV2 dual-volume cell (cup volume of ~1 cm$^3$) at a repetition rate of 6 Hz
and an energy density of approximately 1 J/cm$^2$. The aerosol was transported to a quadrupole ICP-MS
(iCap, Thermo Scientific) on a helium flow at a rate of 0.7 L/min, with 0.4 L/min Argon make-up gas
being added before entering the torch. Monitored masses included $^{23}$Na, $^{24}$Mg, $^{25}$Mg, $^{27}$Al, $^{43}$Ca, $^{44}$Ca,
$^{55}$Mn, $^{88}$Sr and $^{137}$Ba, with one full cycle through the different masses taking 90 ms. Calibration was
performed against a MACS-3 (synthetic calcium carbonate) pressed powder carbonate standard with
$^{43}$Ca as an internal standard. Count rates for the different masses were directly translated into
element/Ca$_{cc}$ (El/Ca$_{cc}$) ratios. Internal precision based on MACS-3 is 4% for Na, 3% for Mg and 4% for
Sr. Accuracy and relative analytical errors, based on measuring international standards JCp-1 coral
(*Porites* sp.) powder and the NIST (National Institute of Standards and Technology) SRM 610 and
SRM 612 (glass) are listed in Table 2. The relatively large offset between the glass standards and the
pressed powders (both MACS-3 and JCp-1) is known not to influence obtained El/Ca$_{cc}$ ratios when
either one is used as calibration standard (Hathorne et al., 2008), but due to the similar matrix, MACS-
3 was chosen as calibration standard.





**Table 2.** Accuracies and precisions for Na, Mg and Sr for the various standards analyzed.

| Standard | n | Accuracy Na (%) | Precision Na (%) | Ac Mg (%) | Pr Mg (%) | Ac Sr (%) | Pr Sr (%) |
|---|---|---|---|---|---|---|---|
| **JCp-1** | 51 | 99 | 6 | 96 | 6 | 96 | 4 |
| **NIST610** | 32 | 119 | 3 | 104 | 2 | 110 | 3 |
| **NIST612** | 29 | 119 | 3 | 104 | 2 | 110 | 2 |


In total, 675 chambers were measured (336 for *Amphistegina* and 339 for *Ammonia*), resulting in
between 52 to 125 single chamber measurements per salinity condition per species. These
measurements were done on the last three (final or F, penultimate or F-1 and F-2) chambers of these
specimens. For *Amphistegina*, these chambers were derived from (condition/no of specimens/average
spots per specimen): S25/28/2.6, S30/40/1.9, S35/60/1.9, S40/27/2 and S45/33/1.4. For *Ammonia*, the
number of analyses were (condition/ no of specimens/ average spots per specimen): S25/44/2.5,
S30/31/1.8, S35/33/1.8, S40/52/1.8, S45/15/1.3. Element concentrations were calculated from the time
(i.e. ablation depth) resolved profiles using an adapted version (for details see Van Dijk et al., 2017a)
of the SILLS (Signal Integration for Laboratory Laser Systems; Guillong et al., 2008)) package for
MATLAB, while taking care to exclude contaminations potentially present on chamber walls
(examples of profile selection: Duenas-Bohorquez et al., 2011; Wit et al., 2013; Mewes et al., 2014;
Mezger et al., 2016; Van Dijk et al., 2017b). Measurements with ablation yields or integrations times
<5 s were excluded from further analysis.
Since there is variability in Ca counts between the laser ablation measurements, single-spot based
Element/Ca$_{cc}$ ratios may cause spurious correlation due to coupled differences in Ca counts. To test
whether observed correlations between Na/Ca$_{cc}$, Sr/Ca$_{cc}$ and Mg/Ca$_{cc}$, based on single-spots, are due to
the use of a common denominator (Ca), we performed a Monte Carlo simulation. In short, the
correlation coefficients between randomly drawn single-spot Mg concentration, divided by measured
Ca, and measured Na/Ca$_{cc}$ concentrations were compared to the correlation coefficient of measured
Na/Ca$_{cc}$ and Mg/Ca$_{cc}$ concentration ratios in our dataset. By using a Kernel fit of the measured data set
to draw the random data set and using the measured Ca as a common denominator we effectively
simulate the spurious correlation. This was repeated 10,000 times and repeated for the couples
Mg/Ca$_{cc}$-Sr/Ca$_{cc}$ and Na/Ca$_{cc}$-Sr/Ca$_{cc}$ (Appendix B).
Furthermore, to test whether Sr/Ca$_{cc}$ and Na/Ca$_{cc}$ variability in *A. lessonii* is not caused by variability in
Mg content due to a potential closed sum effect (since high amounts of incorporated Mg cations could
reduce the Ca content of the shell and hence result in apparently elevated Sr/Ca$_{cc}$ and Na/Ca$_{cc}$), we
calculated maximum variability due to the sole effect of Mg-substitution. For *A. lessonii*, variability
(standard deviation) of ±0.09 mmol/mol in Na/Ca$_{cc}$ and ±0.016 mmol/mol in Sr/Ca$_{cc}$ around the mean
could be caused by variability in Mg/Ca$_{cc}$ (assuming Mg substitutes for Ca in the calcite lattice, and Mg
plus Ca approximates 1 mol per mol calcite). This may have influenced the Sr/Ca$_{cc}$ and Na/Ca$_{cc}$
regression slopes over salinity and also the calculated inter-element correlation coefficients, but only
by a maximum of ±1% for both elements, which is considerably lower than the total observed
variability of 16% and 9%, respectively.


**2.3 Electron Microprobe Mapping**
To investigate variation of element distribution across the chamber wall, a number of cultured
specimens were prepared for Electron Microprobe Analysis (EPMA). From each of the five salinity
conditions, six specimens from both species were selected and embedded in resin (Araldite 2020) in an
aluminum ring (diameter 1 cm) in a vacuum chamber. Samples were polished with a final polishing
step using a diamond emulsion with grains of 0.04 μm. This procedure resulted in exposure of a cross-
section of the foraminiferal chamber wall from which areas for EPMA mapping were selected
(Appendix A). These areas were selected for being perpendicular to the shell outer surface, resulting in
pores completely crossing the exposed chamber wall. Elemental distributions were mapped in
chambers prior to F-3 to study the element distribution across the various layers of calcite (lamella)
produced with the addition of each new chamber. Elemental distribution in the shell wall was measured
using a field emission Electron Probe Micro Analyser (JEOL JXA-8530F HyperProbe) at 7.0kV with a
dwell time of 350 ms, using a spot diameter of 80 nm and a step size between 0.1538 μm and 0.4072
μm (130 x 130 pixels).
Spatial resolution of the EPMA mapping was determined using the software package CASINO (monte
CArlo SImulation of electroN trajectory in SOlids, v 2.48). With the input parameters identical as used
in our analysis (80 nm spot size, beam current 7 KeV, etc.), the simulated surface radius of the
backscattered electrons (i.e. the spatial resolution) equals 590 nm. Semi-quantitative El/Ca$_{cc}$ profiles
were calculated by averaging the El/Ca$_{cc}$ intensities parallel to the banding direction and applying a
constant calibration factor obtained from LA-ICP-MS measurements on the same specimen, similar to
the procedure of Eggins et al. (2004). We did not use the depth-resolved laser ablation-profiles for this
purpose, but used the average value from the profiles for correlation to the EPMA-derived intensities.
**3. Results**
**3.1 Foraminiferal calcite element ratios and partitioning coefficients as a function of salinity**
Per treatment, from lowest to highest salinity, average Na/Ca$_{cc}$ of the newly formed calcite varied
between 9.3-10.8 mmol/mol for *A. lessonii* and 4.7-6.4 mmol/mol (highest salinity) for *A. tepida* (Fig.
1), with a corresponding partition coefficient (note that partition coefficients are 'apparent', not taking
into account speciation/activity of Na) ranging from $1.90*10^{-4}$ to $2.20*10^{-4}$ and from $0.97*10^{-4}$ to
$1.30*10^{-4}$ for *Amphistegina* and *Ammonia*, respectively (Table 3). For both species, sets of single-
specimen Na/Ca$_{cc}$ show slightly skewed distributions towards higher Na/Ca$_{cc}$ for all salinities
(Kolmogorov- Smirnov test, at the 95% confidence level). Combining all specimens (based on the
average of single-spot measurements per specimen), Na/Ca$_{cc}$ shows a positive linear relationship with
salinity for both *A. lessonii* and *A. tepida* (Na/Ca$_{cc}$ = 0.077 ± 0.017 * S + 7.13 ± 0.60, $F_{1,186}$ = 20.9, p <
0.001 for *A. lessonii* and Na/Ca$_{cc}$ = 0.064 ± 0.013 * S + 3.29 ± 0.44, $F_{1,172}$ = 25.9, p < 0.001 for *A.
tepida*, Fig. 1). The observed average relative standard deviation between specimens in Na/Ca$_{cc}$ at each
of the 5 salinities is 15% for *A. lessonii* and 20% for *A. tepida*. The variance in Na/Ca$_{cc}$ between
individual specimens explained by salinity is $\eta^2$=0.08 for *A. lessonii* and $\eta^2$=0.14 for *A. tepida*.





Specimen's average Mg/Ca$_{cc}$ and Sr/Ca$_{cc}$ correlate positively with salinity in *A. tepida* (Mg/Ca$_{cc}$ =
$0.060 \pm 0.011 * S + 0.51 \pm 0.38$ $F_{1, 172} = 29.9$ $p < 0.001$ and Sr/Ca$_{cc}$ = $0.014 \pm 12 * 10^{-4} * S + 1.00 \pm$
$0.04$, $F_{1, 337} = 254$, $p < 0.001$), whereas they do not correlate with salinity in *A. lessonii*. Average
relative standard deviations for the 5 salinity conditions per element are 27% for Mg/Ca$_{cc}$ and 9% for
Sr/Ca$_{cc}$ in *A. lessonii* and 32% in Mg/Ca$_{cc}$ and 7% for Sr/Ca$_{cc}$ for *A. tepida*. In *A. lessonii*, the
proportion of variance in Sr/Ca$_{cc}$ explained by salinity is $\eta^2=0.04$ (p<0.01) (Mg/Ca$_{cc}$ not significant)
and for *A. tepida*, the proportion of variance in Sr/Ca$_{cc}$ explained by salinity is $\eta^2=0.44$ and in Mg/Ca$_{cc}$
$\eta^2=0.19$ (p<0.001).
Single-spot analyses on *Ammonia tepida* show that Na/Ca$_{cc}$ and Mg/Ca$_{cc}$ are significantly correlated
within the salinity treatments, except for condition S=30 (Fig. 3). For the individual salinity treatments,
single-spot Sr/Ca$_{cc}$ and Mg/Ca$_{cc}$, as well as Na/Ca$_{cc}$ and Sr/Ca$_{cc}$ are not correlated significantly with
each other, except for S=25. Between salinity treatments, distributions in this species shift towards
higher Na/Ca$_{cc}$, Sr/Ca$_{cc}$ and Mg/Ca$_{cc}$ values with increasing salinity, although for the range between 30-
40 Na/Ca$_{cc}$ distributions remain rather similar (Fig. 3). For *Amphistegina lessonii*, distributions of
Sr/Ca$_{cc}$ and Mg/Ca$_{cc}$ ratios overlap largely between salinities, and only Na/Ca$_{cc}$ distributions shift
towards higher values (Fig. 3). Within each salinity condition however, single-spot Na/Ca$_{cc}$, Mg/Ca$_{cc}$
and Sr/Ca$_{cc}$ in this species are positively correlated amongst each other, whereby the Na/Ca$_{cc}$ intercept
of these relationships increases with increasing salinity (Fig. 3 and Appendix C).
**Table 3.** Average El/Ca$_{cc}$ ratios of the foraminiferal calcite ±standard error and corresponding apparent
partitioning coefficients, defined as $D_{El}=\dfrac{\frac{El}{Ca} calcite}{El/Ca\ SW}$.

| Sal | n | Na/Ca$_{cc}$ mmol/mol | D$_{Na}$ | Mg/Ca$_{cc}$ mmol/mol | D$_{Mg}$ | Sr/Ca$_{cc}$ mmol/mol | D$_{Sr}$ |
|-----|---|-----------------------|----------|-----------------------|----------|-----------------------|----------|
| *A.l.* | | | | | | | |
| S25 | 65 | 9.29±0.27 | $1.90*10^{-4}$ | 33.35±1.20 | $5.94*10^{-3}$ | 1.80±0.026 | 0.199 |
| S30 | 74 | 9.47±0.21 | $1.90*10^{-4}$ | 32.10±1.20 | $5.64*10^{-3}$ | 1.74±0.020 | 0.189 |
| S35 | 103 | 9.63±0.18 | $1.98*10^{-4}$ | 32.71±1.07 | $5.94*10^{-3}$ | 1.76±0.018 | 0.191 |
| S40 | 50 | 10.25±0.31 | $2.11*10^{-4}$ | 35.22±2.60 | $6.27*10^{-3}$ | 1.74±0.034 | 0.184 |
| S45 | 44 | 10.78±0.30 | $2.20*10^{-4}$ | 33.80±1.68 | $5.90*10^{-3}$ | 1.82±0.036 | 0.189 |
| | | | | | | | |
| *A.t.* | | | | | | | |
| S25 | 109 | 4.75±0.11 | $0.97*10^{-4}$ | 1.90±0.06 | $3.40*10^{-4}$ | 1.34±0.016 | 0.148 |
| S30 | 58 | 5.63±0.22 | $1.13*10^{-4}$ | 2.41±0.09 | $4.24*10^{-4}$ | 1.44±0.013 | 0.156 |
| S35 | 59 | 5.58±0.19 | $1.15*10^{-4}$ | 2.85±0.24 | $5.17*10^{-4}$ | 1.50±0.012 | 0.163 |
| S40 | 93 | 5.70±0.16 | $1.17*10^{-4}$ | 2.73±0.15 | $4.86*10^{-4}$ | 1.55±0.017 | 0.164 |
| S45 | 20 | 6.39±0.37 | $1.31*10^{-4}$ | 3.27±0.27 | $5.70*10^{-4}$ | 1.61±0.038 | 0.168 |

**3.2 Size and chamber effect on Na/Ca$_{cc}$ and inter-specimen variance**
Specimens of *A. lessonii* produced most new chambers at salinities of 25, 30 and 35, closest to the
salinity in their "natural" habitat (Burgers Zoo aquarium, salinity (33.9-34.3; Ernst et al., 2011). Size
averages are not significantly different between these salinity treatments, based on a Kruskal-Wallis
test, whereas specimens grown at salinities 40 and 45 were significantly smaller than those from lower
salinities, reflecting lower chamber addition rates over the course of the culturing experiment at higher



salinity (Fig. 2). Combining al specimens, Na/Ca$_{cc}$ is not significantly related to size in *A. lessonii.*
Specimens of *A. tepida* produced less chambers at salinity 45, possibly because although this species is
used to relatively large salinity shifts in their tidal flat habitat, such a high salinity is probably close to
its tolerance. The lower salinity groups (25, 30, 35) produced larger specimens than the highest
salinities (Fig. 2). Combining al specimens, Na/Ca$_{cc}$ is significantly related to size in *A. tepida,* yet with
a small slope (-0.003) and just within the 95% confidence interval (p=0.04).
Within each salinity tested, single-chambered Na/Ca$_{cc}$ is slightly positively related to size for the
specimens of *A. lessonii* cultured at salinities 25 (slope = 0.008, p < 0.01), 30 (slope=0.002, P<0.05 and
35 (slope=0.005, p<0.001). For the same species, Mg/Ca$_{cc}$ is positively correlated to size at salinities
25, 30 and 35, with a similar slope of 0.03 (p < 0.05). Sr/Ca$_{cc}$ also shows a positive relationship to size
within salinities 25, 30 and 35 with slopes of 0.0007, 0.0003, 0.0005 (p<0.001) respectively. For *A.*
*tepida,* there is only a slight negative correlation between size and Sr/Ca$_{cc}$ for specimens cultured at
salinity 25 (slope=9.9*10-4, p<0.001) and no significant correlation for the other conditions, or
between size and Na/Ca$_{cc}$ and Mg/Ca$_{cc}$ in all salinity groups.
At the lowest salinity, Na/Ca$_{cc}$ in the F-chamber (newest chamber) show slight (0.9 mmol/mol Na/Ca
higher median) but significant higher values than the F-2 chambers for *A. lessonii* (multicompare test
based on Kruskal-Wallis test, p<0.05). For specimens of *A. lessonii* cultured at other salinities and for
*A. tepida* at any of the salinities tested, there no significant correlations between Na/Ca$_{cc}$ and chamber
position were observed (note that only chamber positions F to F-2 were taken into account, as for the
lower chamber position sample numbers were insufficient). Furthermore, chamber position shows no
significant effect on Mg/Ca$_{cc}$ and Sr/Ca$_{cc}$.
To further investigate the variance between and within individuals, a multiway ANOVA was
performed to investigate the effect on Na/Ca$_{cc}$ per salinity condition. Inter-individual variance is
significant and larger than the variance between chamber groups and intra-individual variance in all
salinity groups, with the between individual variability accounting for η$^2$= 0.75 ± 0.11/0.84±0.03 of the
variance (p<0.001) for *A. lessonii* and *A. tepida* respectively. The variance due to chamber position is
not significant, and the remaining intra-individual variance accounts for η$^2$ = 0.09±0.05/0.08±0.05 for
*A. lessonii* and *A. tepida* respectively.

**3.3 Elemental distributions in the chamber wall**

EPMA maps of cross-sectioned chamber walls of *A. lessonii* show, within the resolution limits of the
technique, that bands of elevated Na/Ca$_{cc}$ intensities overlap with zones of elevated Mg/Ca$_{cc}$ (Fig. 6 and
appendix D). Mg bands show a higher amplitude than Na bands, but clearly coincide spatially.
Comparing EPMA maps with the backscatter SEM image of the exposed sections shows that the bands
with the highest Na/Ca$_{cc}$ and Mg/Ca$_{cc}$ occur in the proximity of the primary organic sheet (Fig. 4), with
a number of high Na- and Mg-rich bands with slightly lower maximum intensities occurring towards
the outer chamber surface coinciding with subsequent organic linings. For *A.tepida*, one band of
elevated Mg/Ca$_{cc}$ band is visible coinciding with the POS with no clear Na/Ca$_{cc}$ banding being
detected.



## 4. Discussion

### 4.1 The effect of salinity and DIC on Na/Ca$_{cc}$, Mg/Ca$_{cc}$ and Sr/Ca$_{cc}$

The Na/Ca$_{cc}$ single-specimen data of the cultured *A. lessonii* and *A. tepida* both correlate positively with salinity (Table 3, Fig. 1). This is in line with previous calibrations (for *Ammonia tepida*; Wit et al., 2013, for cultured *Globigerinoides ruber*; (Allen et al., 2016) and for field-collected *G. ruber* and *G. sacculifer*; Mezger et al. (2016)). However, our Na/Ca-salinity calibration for *A. tepida* is somewhat less sensitive than that observed earlier for the same species (Wit et al., 2013). An offset in Na/Ca$_{cc}$ values between calibrations for a single species has been reported before (e.g. Mezger et al., 2016 and Allen et al., 2016 for the planktonic *G. ruber* and *G. sacculifer*). Such an apparent discrepancy between studies may be caused by differences in one of the not targeted conditions between cultures or in situ conditions (e.g. carbon chemistry, light intensity). Alternatively, subtle analytical differences (e.g. differences in cleaning procedures), statistical reasons (for example differences in the number of analyses or sample size) or the effect of genotypic variability on element incorporation (Sadekov et al., 2016) may also play a role. Although the calibration presented here consists of much more data points compared to those in Wit et al. (2013), we do not want to dismiss the latter as several parameters (like cleaning procedures) inevitably might not have been identical. As such the difference observed between studies merely illustrates the potential range for this species.

Contrasts in sensitivities such as observed for Na/Ca$_{cc}$ between calibrations also apply to Mg/Ca$_{cc}$ and Sr/Ca$_{cc}$, both of which here show an increase with salinity in *A. tepida* but not in *A. lessonii* (Fig. 1). Previous culturing experiments with *Ammonia tepida*, however, showed a smaller sensitivity of Mg/Ca$_{cc}$ to salinity (0.029-0.0044 mmol/mol change per salinity unit; Dissard et al., 2010) than that reported here (0.06). Still, all these sensitivities are considerably lower than that reported in Kisakürek et al. (2008) for the planktonic *G. ruber* (0.23 when Mg/Ca$_{cc}$ is assumed to increase linearly with salinity), but in the same range as that reported by Nürnberg et al. (1996) for *G. sacculifer* (0.05). The sensitivity of Sr/Ca$_{cc}$ to salinity in *A. tepida* (0.014; Table 3) is comparable to that for *O. universa* (0.008; Lea et al., 2008), *G. ruber* (0.02; Kisakürek et al., 2008) and similar to the significant effect of salinity on Sr incorporation in the same species (0.01-0.02, depending on temperature) found by Dissard et al. (2010).

Seawater carbonate chemistry is an additional factor potentially affecting trace metal uptake (e.g. Lea et al., 1999; Keul et al., 2017; Russell et al., 2004). Since salinity and dissolved inorganic carbon concentration in the culture media co-varied in our experiments similar to the natural environment (Table 1), Na/Ca$_{cc}$ in our cultured specimens also correlates positively to seawater [DIC]. However, sodium incorporation has been shown to be independent from changes in carbonate chemistry in cultured *Amphistegina gibbosa* and several other benthic hyaline and porcelaneous species (Van Dijk et al., 2017a). Additionally, Allen et al., (2016) also found no significant effect of carbonate chemistry (i.e. varying [CO$_3^{2-}$]) on Na incorporation in cultured *G. ruber*, suggesting that the variability in Na/Ca$_{cc}$ observed here in *A. lessonii* can be attributed to changes in salinity rather than [DIC]. Previous studies showed that Sr/Ca$_{cc}$ correlates positively to [DIC] in *A. tepida* (Keul et al., 2017), which may account for part of the correlation between Sr/Ca$_{cc}$ and salinity reported here for this species. The

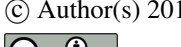



published sensitivity of Sr/Ca$_{cc}$ to [DIC] is approximately $2*10^{-5}$ mmol/mol change in Sr/Ca$_{cc}$ for every
1 µmol/kg change in [DIC], likely representing the maximum potential effect of DIC on Sr partitioning
given that others found no significant effect (Dissard et al., 2010a). For a change in ~850 µmol/kg
(Table 1), this would amount to an increase in Sr/Ca$_{cc}$ of 0.019 mmol/mol (Keul et al., 2017) over the
salinity range studied here, thereby accounting for approximately 7% of the total observed change in
Sr/Ca$_{cc}$ (Table 3). Inorganic carbon chemistry is known to affect growth rates and shell weights in
benthic foraminifera (Dissard et al., 2010a; Keul et al., 2013), which in turn, may affect incorporation
of Sr and Mg, hence providing a mechanistic link between inorganic carbon chemistry and element
partitioning.
The absence of an (strong) impact of DIC on Mg/Ca$_{cc}$ in foraminiferal calcite (our results; Fig. 1;
Kısakürek et al., 2008; Dissard et al., 2010a; Russell et al., 2004) implies that changes in combined
Mg/Ca$_{cc}$ and Na/Ca$_{cc}$ in low-Mg foraminiferal species can be used to reconstruct salinity and improve
temperature estimates. Any additional changes in the marine inorganic carbon system will have a much
larger impact on other elements (e.g. B, Zn, U), so that the combined analyses of all these elements will
allow for a complete reconstruction of past seawater conditions.
El/Ca ratios of specimens of both species grown within each salinity condition are characterized by a
relatively large variability. Of the overall data set salinity only explains around 8% of the variation in
Na incorporation for *A. lessonii* and 14% 19% and 44% of Na, Mg and Sr incorporation. However, for
*A. lessonii*, the mean values (which translates to the values obtained from traditional solution-ICP-MS)
fit the regression model relatively well (Fig. 1). However, given the low sensitivity, many specimens
are required to reduce the uncertainty (Appendix E). This is reflected by the relatively wide prediction
bounds for the Na/Ca-salinity regressions, indicating an uncertainty associated with a single Na/Ca$_{cc}$
measurement. The relatively inter-specimen variability in element/Ca$_{cc}$ ratios has been reported and
discussed before (e.g. Sadekov et al., 2008; De Nooijer et al., 2014a), but the cause for this variability
remains to be identified.

### 4.2 El/Ca$_{cc}$ variability at the inter-specimen and inter-species level

Single-chamber measurements show that Na/Ca$_{cc}$ for both species varies between chambers (i.e.
specimens) with a RSD of 15%-20%, despite identical culture conditions (Fig. 1). Since the analytical
error on Na/Ca$_{cc}$ accounts for approximately 2% (Table 2), a large portion of the observed variability
between specimens must be due to ontogeny and/or inter-specimen differences in biomineralization
controls (De Nooijer et al., 2014).
Foraminiferal shell size at salinities 40 and 45 are significantly smaller than those cultured at lower
salinities. When combining data from all salinities, however, there is no (*A. lessonii*) or only a very
small (*A. tepida*) negative correlation between Na/Ca$_{cc}$ and shell size, as suggested earlier by Wit et al.
(2013). Potentially the earlier observed co-variation was caused by an indirect co-variation rather than
a causal relationship. Also within treatments, a relationship between Na/Ca$_{cc}$ and size is either opposite
(i.e. positive) or absent. Hence, size is unlikely to be responsible for any of the observed inter-specimen
variability in Na/Ca$_{cc}$, which is supported by the absence of a correlation between chamber position
(and hence ontogenetic stage) and Na/Ca$_{cc}$. This implies that differences in Na/Ca$_{cc}$ between chambers

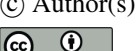


do not need to be taken into account when applying Na/Ca$_{cc}$ as a proxy, although the large inter-
specimen variance in Na/Ca$_{cc}$ requires sufficient specimens (n>30, for an error margin <5% at the 95%
confidence level; Sadekov et al., 2008; De Nooijer et al., 2014a) to be analyzed. As most variability is
between individuals rather than between chambers (section 3.3), analyzing more chambers of the same
specimen does not necessarily improve the precision of the salinity estimate. Without a major effect of
ontogeny, physiological processes at the organismal level are more likely to cause observed large inter-
specimen variability in Na/Ca$_{cc}$.
In *A. lessonii*, single-spot Na/Ca$_{cc}$, Sr/Ca$_{cc}$ and Mg/Ca$_{cc}$ are correlated amongst each other within each
salinity condition (Fig. 3). Correlation coefficients between the three element ratios are similar for the
different salinities, with superimposed an increase in the Na/Ca$_{cc}$ relative to that of Mg/Ca$_{cc}$ and Sr/Ca$_{cc}$
with increasing salinity (Appendix C). In contrast, single-spot Sr/Ca$_{cc}$ and Mg/Ca$_{cc}$ in *A. tepida* are not
correlated, whereas incorporation of all these elements increases significantly with salinity. Within
salinities Mg/Ca$_{cc}$ and Na/Ca$_{cc}$ are significantly correlated in 4 out of the 5 salinities, but with much
lower correlation coefficients compared to *A. lessonii* (Fig. 3 and Appendix C). However, between the
different salinities these elements are correlated in *A. tepida*, implying that for *A. tepida* salinity is one
of the actual parameters controlling element uptake.

375       The differences in (an absence of) a correlation between elements between the two species

studied here likely reflect differences in their calcification pathways. At the same time, such a
difference may also explain why Sr/Ca$_{cc}$ and Mg/Ca$_{cc}$ are correlated to salinity in *A. tepida*, but not in
*A. lessonii* (4.1). The overall element composition of the calcite precipitated by *A. lessonii* suggests that
the calcification process of this species may more closely resemble inorganic calcite precipitation,
compared to *Ammonia tepida* and other low-Mg calcite precipitating species. In the intermediate-Mg
calcite species, crystal lattice strain is elevated which may promote incorporation of other elements
through stress compensation (Mucci and Morse, 1983; Mewes et al., 2015). This would explain the
observed inter-element correlations within salinities. Another difference between the species studied
here may be caused by differences in CaCO$_3$ phase shifts during calcite precipitation (e.g. Bots et al.,
2012; De Yoreo et al., 2015). A metastable vaterite pre-cursor phase recently found in two planktonic
species may explain the low Mg incorporation relative to inorganic calcite (Jacob et al., 2017). The
higher Mg contents of *A. lessonii* could be related to the (partial) absence of a vaterite-calcite
transformation in this species. A higher Mg concentration at the site of calcification might result in a
phase shift from amorphous calcium carbonate (ACC) directly into to calcite, with Mg stabilising the
ACC, as described by Littlewood et al. (2017). The absence of a vaterite precursor phases also
enhances the incorporation of other metals incompatible to calcite, such as Sr (Littlewood et al, 2017)
and hence may contribute to the inter-species differences in element partitioning similar to that
observed here. Although the strong fractionation against Mg in *A. tepida* could reflect double
fractionation through a vaterite-calcite transformation (Jacob et al., 2017) the low-Mg content might as
well reflect a more enclosed site of calcification, whereby ions are mainly transported trans-membrane
(Nehrke et al., 2013), but the experiments here do not allow distinguishing these two potential
mechanisms. Trans-membrane transport (TMT) of Ca$^{2+}$ and concomitant leakage of Mg$^{2+}$ and Sr$^{2+}$
might be more sensitive to differences in ionic strength and element concentrations, hence possibly





explaining the salinity effect on the incorporation of these elements in *A. tepida* whereas it does not in
*A. lessonii,* assuming that TMT relatively contributes more to the supply of ions to the site of
calcification in this species compared to *A. lessonii*, which might be relatively more dependent on
seawater vacuolisation.

**4.3 Intra-specimen variability**

In both species, Mg is found to be elevated in bands located close to the primary organic sheet and
(often less pronounced) to other organic layers (Fig. 4), present in rotaliid species due to their lamellar
calcification mode (Reiss, 1957, 1960). This is similar to reports of within-chamber wall banding in
many elements in other species (Branson et al., 2016; Eggins et al., 2004; Sadekov et al., 2005; Paris et
al., 2014; Spero et al., 2015; Fehrenbacher et al., 2017; Kunioka et al., 2006; Steinhardt et al., 2015;
Hathorne et al., 2009). As in other studies, the Na- and Mg- bands are spatially correlated (Fig. 4). For
*Ammonia tepida*, the banding in both elements is less pronounced than for *Amphistegina lessonii*,
which is not surprising given the higher average $El/Ca_{cc}$ ratios in the latter species. This inter-species
difference observed in the Mg- and Na-maps implies that the concentration of Mg and Na within the
high concentration band is lower in *A. tepida* than in *A. lessonii*. Alternatively, as the observations are
close to the spatial resolution of the method, the observed pattern could also be due to the band's width
being smaller in *A. tepida* compared to *A. lessonii*. When comparing the distribution of the two
elements within one specimen, the $Mg/Ca_{cc}$ bands are more pronounced than those of $Na/Ca_{cc}$,
particularly for *A. lessonii* (Fig. 4).
The spatial correlation between the intra-shell distributions Mg and Na suggests a coupled control on
these elements during the calcification process, which is in line with the observed inter-specimen
correlations. This suggests that the incorporation of these cations is influenced by similar
biomineralization mechanisms, related to seawater vacuolization (Erez, 2003; Bentov and Erez, 2006),
trans-membrane transport of elements (Nehrke et al., 2013) and/or metastable precursor phases (Jacob
et al., 2017). The relative contributions of these mechanisms might differ between species, resulting in
the observed differences in element incorporation between species. Differences in the efficiency of
such processes between specimens might cause the observed inter-specimen variability, whereas
changes in these processes during the calcification time could be responsible for the observed
correlation between elements within the chamber wall.

**5. Conclusions**

By extending existing calibrations of the $Na/Ca_{cc}$-salinity proxy to the intermediate-Mg calcite
precipitating benthic foraminifer *Amphistegina lessonii*, we show that the $Na/Ca_{cc}$ increase as a
function of salinity is similar to that in previously studied species. The absolute $Na/Ca_{cc}$ for *A. lessonii*
is, however, higher than that in *Ammonia tepida*. In *A. tepida*, $Mg/Ca_{cc}$ and $Sr/Ca_{cc}$ are positively
correlated to salinity, whereas they are not impacted by salinity in *A. lessonii*. Within each salinity,
single chamber-$Na/Ca_{cc}$ and $Mg/Ca_{cc}$ are positively correlated in *A. tepida*, whereas single chamber-
$Sr/Ca_{cc}$ is not correlated to either $Mg/Ca_{cc}$ or Na in this species. For *A. lessonii*, all $Sr/Ca_{cc}$, $Mg/Ca_{cc}$ and



Na/Ca$_{cc}$ combinations are positively correlated at the single chamber level. EPMA mapping of Na and
Mg within chamber walls of cultured specimens shows that in *A. lessonii*, Na/Ca$_{cc}$ and Mg/Ca$_{cc}$ occur
in elevated bands in close proximity to the primary organic lining. For specimens of *A. tepida*, Mg-
banding appears similar to that in *A. lessonii*, whereas Na-banding is less prominent in this species.
**Acknowledgements**
We would like to thank Wim Boer for assistance with LA-ICP-MS measurements, Patrick Laan for
seawater measurements and Karel Bakker for DIC measurements. We kindly thank Max Janse
(Burgers Zoo, Arnhem) for providing stock specimens of *A. lessonii* and Kirsten Kooijmans (NIOZ)
for providing cultures of *Dunaliella salina*. Sergei Matveev is thanked for assistance with the Electron
Microprobe analysis and Leonard Bik for assistance with polishing the samples. This work was carried
out under the program of the Netherlands Earth System Science Centre (NESSC), financially supported
by the Ministry of Education, Culture and Science (OCW) (Grantnr. 024.002.001) and Darwin Centre
for Biogeosciences (program 3020).

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





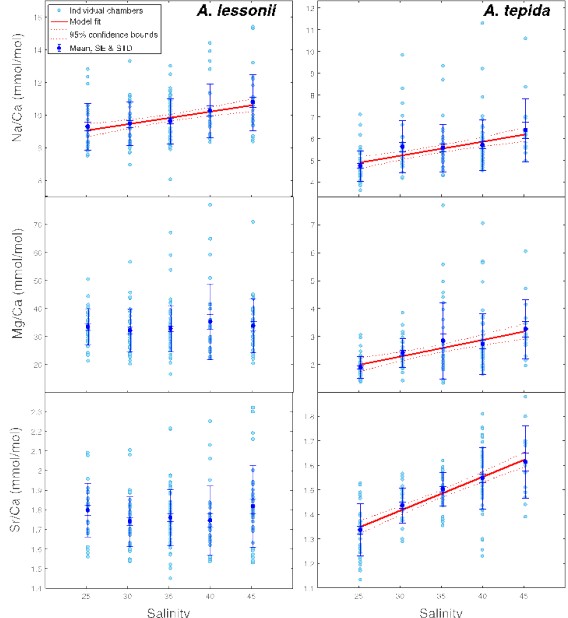


**Figure 1. Foraminiferal Na/Ca$_{cc}$, Mg/Ca$_{cc}$ and Sr/Ca$_{cc}$ versus salinity. Light blue dots represent the average**
**per specimen (n= 359 for A. lessonii, n= for A. tepida, with ±3 measured chambers per individual), dark**
**blue dots indicate the mean, with inner error bars indicating the standard error and outer error bars the**
**standard deviation for each treatment. The linear regression model (red line) is based on the individuals'**
**mean, with the 95% confidence interval of the regression in dashed lines.**

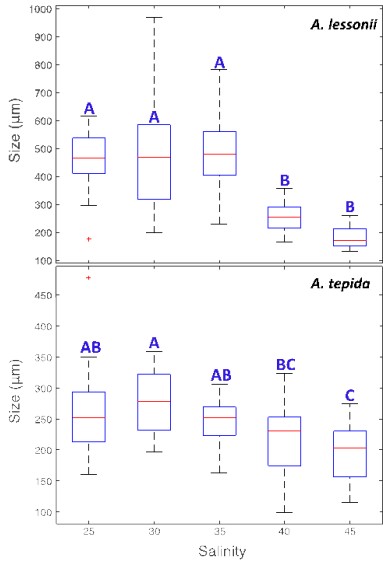

606



**Figure 2. Boxplot showing the size distributions (median, 1ˢᵗ and 3ʳᵈ quartiles, minimum and maximum values) for each salinity condition, n=68, 74, 115, 53, 45 for *A. lessonii* and n= ... for *A. tepida*. Letters indicate significant different population means, based on ANOVA (p<0.001).**

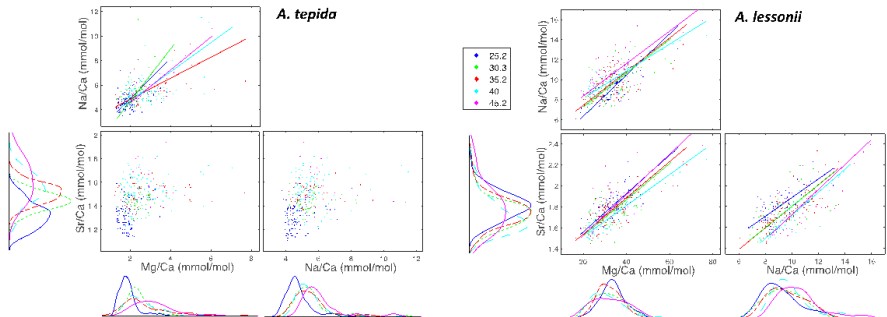

**Figure 3. Individual chamber LA-ICP-MS analyses showing correlations between foraminiferal Mg/Ca$_{cc}$, Sr/Ca$_{cc}$ and Na/Ca$_{cc}$. for *A. tepida* (left) and *A. lessonii* (right) per salinity condition. Significant orthogonal linear regressions for are indicted with a line, colour coded for salinity (see legend). Correlation coefficients, slope and intercepts of these regressions can be found in Appendix C. Within salinity conditions, element ratios are strongly correlated with each other in *A. lessonii*, whereas in *A. tepida*, element ratios do not (strongly) correlate with each other. When combining all single-spot data in *A. tepida*, element ratios correlate amongst each other because the incorporation of all three elements increases with salinity, shifting the distributions to higher values. In *A. lessonii*, only the Na/Ca$_{cc}$ distributions shift towards higher values with increasing salinity, whereas Mg/Ca$_{cc}$ and Sr/Ca$_{cc}$ distributions are relatively similar between salinity conditions.**



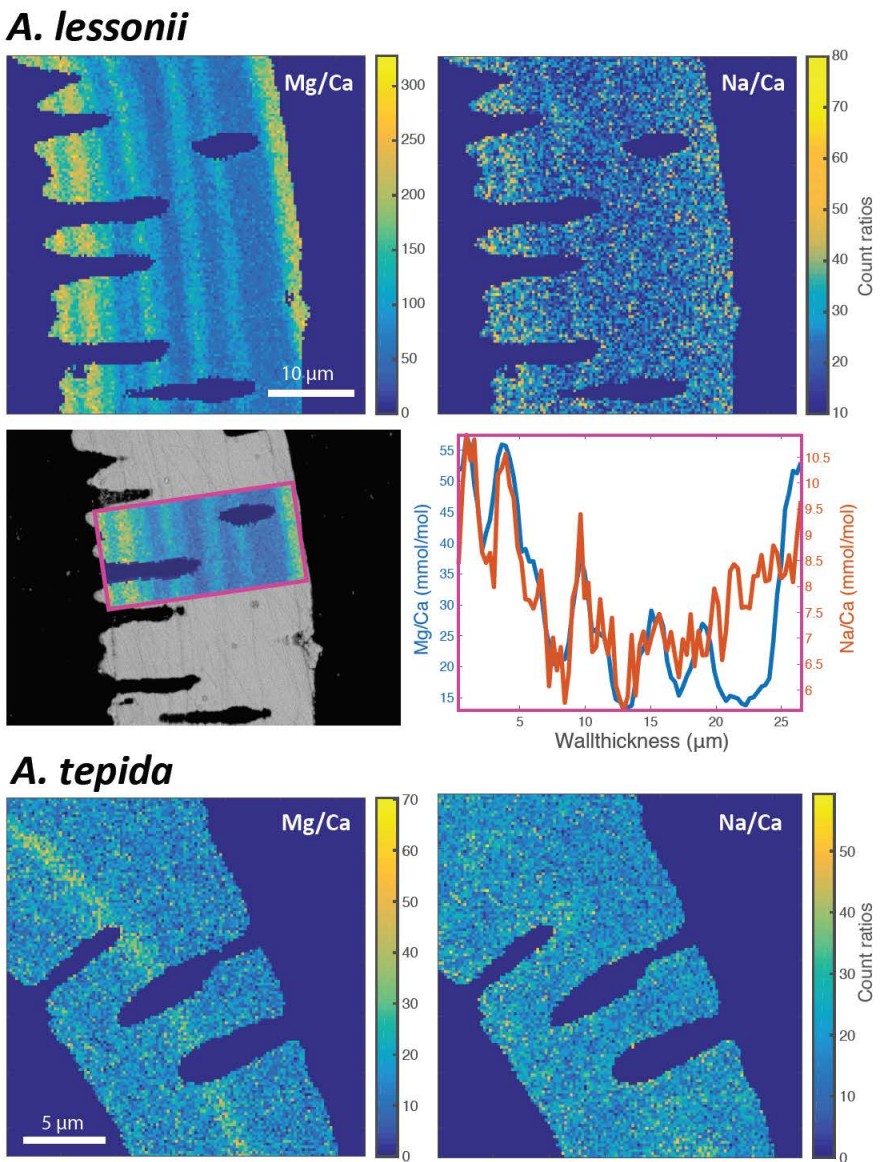

621

**Figure 4. Foraminiferal Mg/Ca$_{cc}$ (upper left) and Na/Ca$_{cc}$ (upper right) intensity ratio maps, obtained with**
**EPMA, for a specimen of _A. lessonii_ grown at a salinity of 30. The lower right panel shows profiles for**
**Mg/Ca (blue) and Na/Ca (red), based on averaged EPMA ratios scaled to LA-ICP-MS measurements of the**
**same specimen, of an averaged transact area through the chamber wall perpendicular to the POS. The**
**transect area is indicated in the lower left panel, on top of a backscatter SEM image, showing that the high**
**El/Ca bands overlap with the primary organic sheet (POS, left) and subsequent organic linings (towards the**
**right). Correlation coefficient R$^2$=0.56 (p<0.001) for Mg versus Na, based on element intensity counts to**
**exclude covariation with Ca. Lower panels show Mg/Ca$_{cc}$ (lower left) and Na/Ca$_{cc}$ (lower right) intensity**



ratio maps, obtained with EPMA, for a specimen of *A. tepida* grown at a salinity of 35. See C for the results
for three more specimen.

**Appendix**

**Appendix A.**

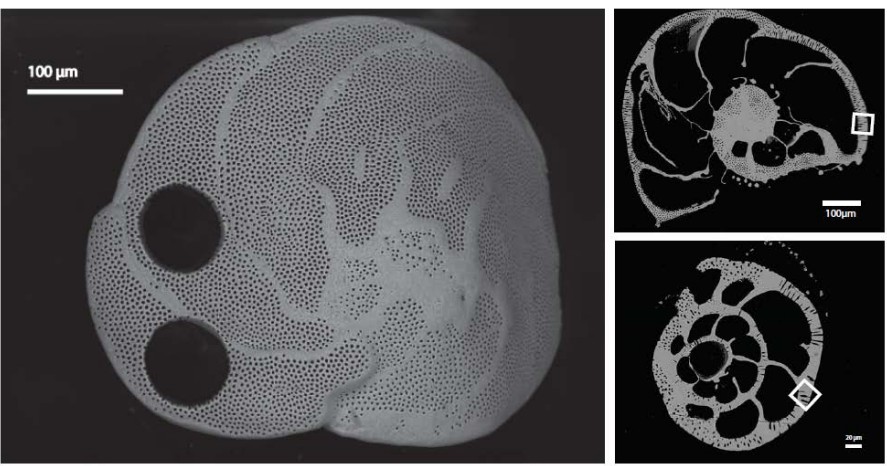

SEM image of a specimen of *A. lessonii* showing LA-ICP-MS measurement spots (left) and SEM images of
specimens of *A. lessonii* (upper right) and *A. tepida* (lower right) embedded in resin and polished for
Electron Probe Micro Analysis, the mapping area is depicted with a white box.

**Appendix B.**

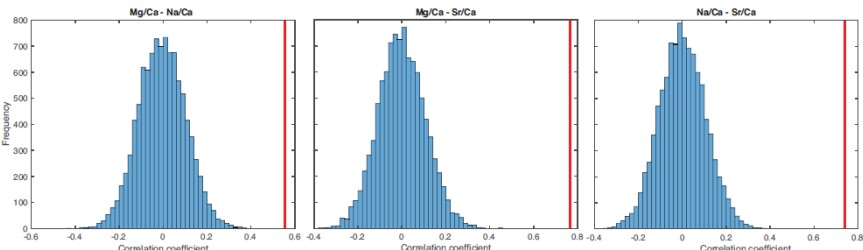

Results of the Monte Carlo analysis showing that the measured correlation coefficients for the inter-
specimen correlations between the measured $El^1/Ca_{cc}$ and $El^2/Ca_{cc}$ are not caused by a spurious correlation
due to the common denominator $Ca_{cc}$, showing that the measured correlation coefficient is significantly
higher then the distribution of the correlation coefficients between 10.000 randomly drawn $El^1$
concentrations/measured Ca concentration and measured $El^2/Ca$ concentrations. This test is based on the
concentration results from a single labbook (measurement run) with specimens of *A. lessonii* cultured at a
salinity of 35.



**Appendix C.**

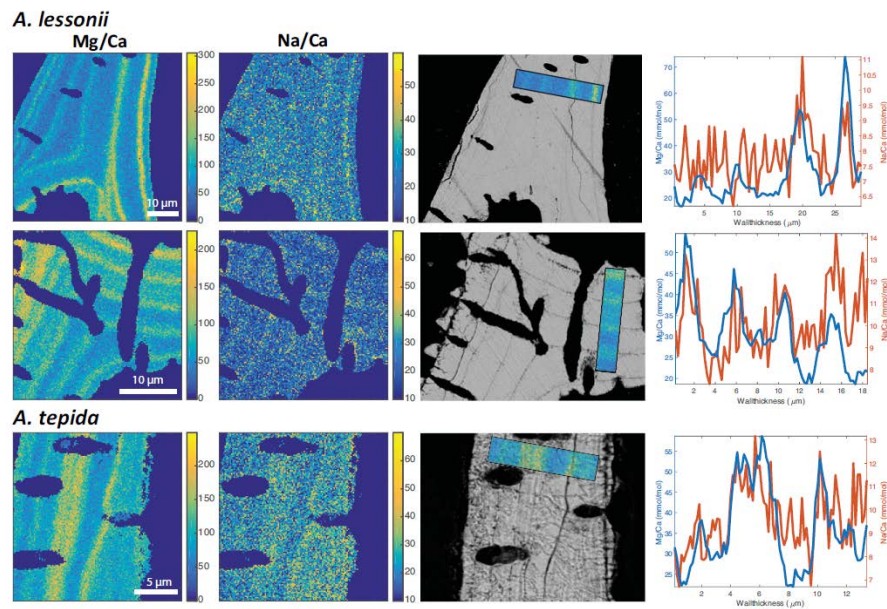


**Foraminiferal Mg/Ca$_{cc}$ and Na/Ca$_{cc}$ (left two panels) intensity ratio maps, obtained with EPMA, for three**
**specimens of *A. lessonii* grown at a salinity of 30 (upper panels), 25 (middle panels) and 40 (lower panels).**
**Right panels shows profiles for Mg/Ca (blue) and Na/Ca (red), based on averaged EPMA ratios scaled to**
**LA-ICP-MS measurements of the same specimens, of an averaged transact area through the chamber wall**
**perpendicular to the POS. The transect areas are indicated on top of backscatter SEM images, showing that**
**the high El/Ca bands overlap with the primary organic sheet (POS) and subsequent organic linings.**

**Appendix D.**





### Orthogonal regression results

| *A. lessonii* | | | | *A. tepida* | | | |
|---|---|---|---|---|---|---|---|
| p-value (x,y) | r | Slope | y-Intercept | p-value (x,y) | r | Slope | y-Intercept |
| **(Mg/Ca, Na/Ca)** | | | | **(Mg/Ca, Na/Ca)** | | | |
| p<0.001 | 0,71 | 0,21 | 2,21 | p<0.001 | 0,35 | 1,64 | 1,63 |
| p<0.001 | 0,52 | 0,17 | 3,89 | p>0.05 | 0,24 | 2,14 | 0,41 |
| p<0.001 | 0,57 | 0,17 | 4,06 | p<0.001 | 0,62 | 0,86 | 3,14 |
| p<0.001 | 0,89 | 0,14 | 5,44 | p<0.001 | 0,53 | 1,19 | 2,35 |
| p<0.001 | 0,69 | 0,16 | 5,12 | p<0.001 | 0,84 | 1,34 | 1,89 |
| **(Mg/Ca, Sr/Ca)** | | | | **(Mg/Ca, Sr/Ca)** | | | |
| p<0.001 | 0,63 | 0,02 | 1,19 | p<0.01 | 0,28 | 0,23 | 0,89 |
| p<0.001 | 0,64 | 0,02 | 1,18 | p>0.05 | 0,20 | 0,13 | 1,13 |
| p<0.001 | 0,76 | 0,02 | 1,19 | p>0.05 | 0,18 | 0,06 | 1,33 |
| p<0.001 | 0,90 | 0,01 | 1,22 | p>0.05 | 0,17 | 0,13 | 1,19 |
| p<0.001 | 0,73 | 0,02 | 1,14 | p>0.05 | -0,28 | -0,16 | 2,12 |
| **(Na/Ca, Sr/Ca)** | | | | **(Na/Ca, Sr/Ca)** | | | |
| p<0.001 | 0,58 | 0,09 | 1,00 | p<0.01 | 0,29 | 0,14 | 0,66 |
| p<0.001 | 0,47 | 0,10 | 0,78 | p>0.05 | 0,10 | 0,06 | 1,10 |
| p<0.001 | 0,72 | 0,10 | 0,77 | p>0.05 | 0,23 | 0,07 | 1,10 |
| p<0.001 | 0,94 | 0,11 | 0,62 | p>0.05 | 0,18 | 0,11 | 0,94 |
| p<0.001 | 0,80 | 0,12 | 0,53 | p>0.05 | -0,32 | -0,12 | 2,34 |


**Results for the orthogonal regressions testing the correlations between single-spot El[1]/Ca and El[2]/Ca, within**
**each salinity conditions, for** *A. lessonii* **and** *A. tepida***.**

**Appendix E.**



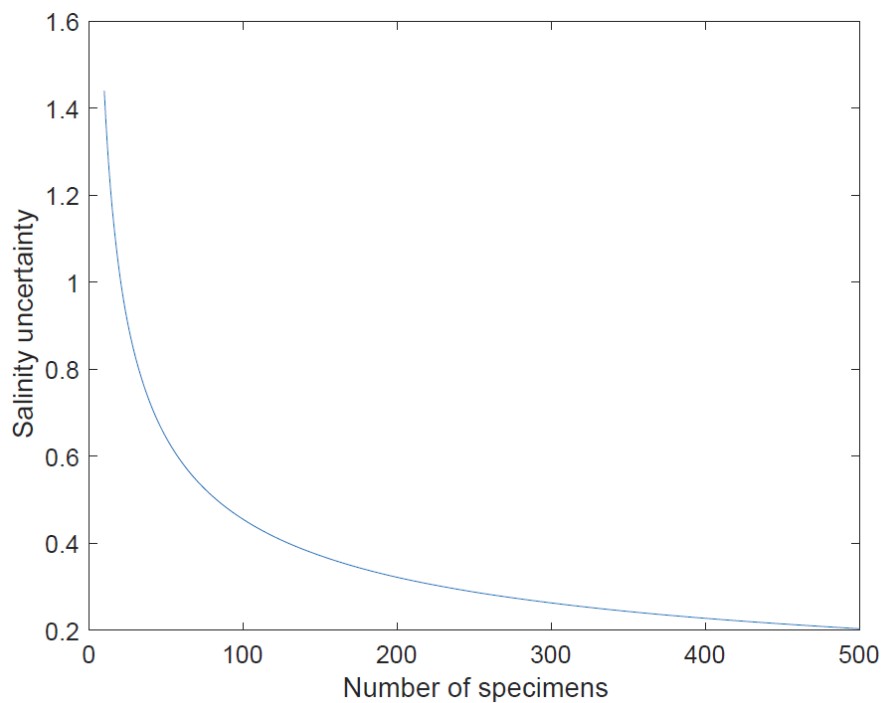


**Figure showing the relationship between the salinity uncertainty and number of measured specimens for the**

**Na/Ca$_{cc}$ - salinity calibration of *A. lessonii*, calculated following Eq. (1):**

**Salinity uncertainty=(2\*RSD\*Number of specimens$^{-0.5}$)/Sensitivity** **(1)**

**Whereby sensitivity is the slope of the calibration.**

669