# Peer review of "benthic foraminiferal species with contrasting"

_Biogeosciences, 2017_

## Referee Comment (RC1) · Anonymous Referee #2 · 23 Jan 2018

General comments

A paper by Geerken et al. reported variabilities in incorporation ratios of Na and other elements to Ca under different salinity levels, observed in various scales of geochemical anlalyses (intra-chamber, inter-chamber, inter-specimen, and inter-species levels). Overall, this paper is well-written, well-considered, and well-structured. The results would be useful for paleo-salinity reconstruction. However, I suggest that the authors consider the following criticisms and comments to improve the final version of this paper.

The main criticism of this paper is a high inter-specimen variability of incorporation

ratios despite using asexually reproduced juveniles incubated in the same petri dish. The authors explained this was caused by differences in the efficiency of calcification processes between specimens (L424-425). If so, what caused the efficiency of calcification process of the specimens? Physiological conditions of each specimen (nutrition, health, and food availability) and the microenvironment in culturing dishes may be clues to answer this question. These have not been measured in this study, but are partly reflected in growth rates of cultured specimens. I suggest the authors compare a relationship between TE/Ca and growth rates for each specimen, not comparing by their average values. This has been briefly described in L 243-248, but data and figures are not shown.

Another criticism is a covariation between salinity and DIC. In this paper, these two parameters are positively correlated. The authors explained that this phenomenon is similar to the natural environment (L311-312), but this does not always occur in the natural environments (e.g. groundwater-seepage area). The authors could have manipulated the carbonate chemistry of culturing media to keep them constant and to make salinity the only variable. I suggest that the authors discuss the limitation of application for this proxy calibration to field specimens where salinity and DIC have not covaried/negatively covaried.

The Discussion of Sections 4.2 and 4.3 should be combined and conclusions in this paper should be based on combined results of various analytical levels of all data sets including EPMA results. For example, Mg and Na are richly concentrated in organic layers based on EPMA results (Section 4.3). I wonder if this finding might affect your conclusions described in former sections.

Specific comments

L32-33: unclear what the authors mean to say.

L35-39: need references

Section 2.1: Descriptions of two species are mixed up and make me confused. Better to separate if culturing conditions are different. For example, did two-third of all incubated specimens reproduced for both species? Were Ammonia specimens incubated at 25C in spite of their original cold habitat from the Wadden Sea? In addition, culturing protocol should be described after explaining culture media preparation (Section 2.2.1).

L95-98: As far as my understanding, you changed culture media every week, then you put algal food and kept the culture media for one week. In that case, did the water quality (salinity and nutrients) change during a week? How did you seal the petri dishes to avoid the salinity changes?

L99-100: Does organic material removal affect the element ratio because Mg and Na are richly incorporated in POS?

L109: Describe the definition of salinity and how to measure it.

Table 1: Do these values indicate the mean values? Better to show the variations prior to and after changing culture media.

L127: Is a laser spot diameter small enough to measure the single chamber for small Ammonia specimens?

Table 2: Explain Ac and Pr, and add the error. List the result of MACS-3 as well. I do not understand why accuracy is over 100%.

L147-150: List them in Table or Appendix.

L243-248: Data and results are not presented.

L246: need references

L276: How did you identify POS from the SEM/EPMA maps?

L325-328: Do you have any ideas how inorganic carbon chemistry affect growth rates

and shell weights? I think in future you have to more carefully check the physiology of cultured foraminifers to understand the proxy calibration, as described in the L364.

L329-334: I think this paragraph is meaningless, better to delete.

L357-358: I think the latter sentence does not support the former sentence.

Conclusions: This is just the summary of results. What did you think about these results?

Technical corrections

L56 (Allen et al., 2016): not italicized

L117: delete : or (

L120: space between with and 0.14

L224: Fig. 3 should be Fig. 2 in the order of appearance.

Figure3, L612: Results of A. lessonii should be placed on the left side for consistence throughout the paper.

L613: Delete "for" after "regression"

L234: average between inter-specimens?

L235: unclear after =

L243 and 247: replace al to all?

L273: Fig. 6 must be Fig. 4.

L274: Appendix D must be Appendix C.

Table 3: What are A.l. and A.t.?

Fig. 4: should be subdivided by a, b, c, d,. . .

L625: transact > transect, No explanation for POS

L627: POS should be indicated in the EPMA maps.

L284-286: better to write "(for Ammonia tepida, Wit et al., 2013; for cultured Globigeri-noides ruber, Allen et al., 2016; for field-collected G. ruber and G. sacculifer, Mezger et al., 2016)".

L288-289: better to write "... reported previously for the planktonic G. ruber and G. sacculifer (e.g. Mezger et al., 2016; Allen et al., 2016)."

L290: not targeted > non-target

L329: Unclear expression: The absence of an (strong) impact of ...

L329: Fig. 1 is related to this sentence?

L337: comma between 14% and 19%

L347: RSD needs explanation

L375: The difference in (an absence of) a correlation between elements between the two species > The difference or an absence of a correlation among elements between the two species?

---

## Referee Comment (RC2) · Anonymous Referee #3 · 28 Feb 2018

The study by Geerken et al. reports the influence of salinity on Na/Ca, Mg/Ca and Sr/Ca of the benthic foraminiferal species A. lessonii and A. tepida, with a focus on Na/Ca as a possible salinity proxy. Element ratios have been determined by laser ablation ICP-MS and foraminiferal specimens have been collected from cultures with different different salinities. The authors found a significant correlation between salinity and Na/Ca and additionally made elemental mappings of the different elements on cross-sections of selected specimens with electron microprobe.

First of all I have to say that I was impressed to see the amount of data and work which has been put into the analyses and culturing. More than 200 specimens have

been analysed with more than 600 measurements! I regret to say after reading the manuscript carefully that there is a substantial problem with this study and at least major revisions are required for the manuscript. The problem is already mentioned by the authors themselves in the first paragraph of the discussion: The cleaning procedures.

The authors mention differences to former calibrations of Na/Ca as salinity proxy within the same species (A. tepida). Wit et al. (2013) used 5% sodium hypochloride to bleach the forams and remove organic contaminations. Geerken et al. now report that they bleached the forams in 70% $H_2O_2$. I do not think this can be true. Most of the $H_2O_2$ solutions which are commercially available are only 30%. In this case it would be 70% of a 30% solution which would be around 21%. This is still much to high! There are a lot of studies about cleaning procedures for analyses on foraminifera and standard methods have been developed based on studies which started already in the 1980s. The standard method for organic matter removal, which is widely applied in most of the labs, is bleaching in a 1% $H_2O_2$ solution (Barker et al., 2003). Barker et al.(2003) also showed that there is a significant impact of different oxidative treatments on foraminiferal Mg/Ca, though not on Sr/Ca ratios. Why did the authors not apply these standard cleaning techniques?

The SEM pictures within appendix C already show that, especially A. tepida appears to be not very well preserved anymore after the treatment. The EMP mappings shown in figure 4 also indicate chemical alteration on the outer part of the test walls of A lessonii (see Mg/Ca mapping and SEM picture). While most of the standard cleaning procedures are used for analyses of bulk samples with ICP-MS, I am not aware if there is any study on the impact of different cleaning procedures on laser ablation analyses. Are there studies at all on different cleaning procedures regarding microanalyses? Otherwise I would strongly recommend to develop a standard treatment also for laser ablation studies. Only this would ensure comparability between different studies.

From this point of view I think the authors cannot provide a quantitative calibration for

Na/Ca ratios as a salinity proxy. They can show that there is a significant relative effect of salinity on Na/Ca and thus give further evidence for the validation of this proxy. I suggest to rewrite the manuscript bearing this in mind.

Below I listed some additional points of revision:

General:

There are a lot of repetitions in the text, especially in the discussion. The manuscript is quite long and would benefit a lot by shortening some parts of the text.

Line 46 - 50: Nürnberg et al., (1996) was on Mg/Ca and not on Sr/Ca as far as I know.

Page 71 - 74: Please rewrite. This sentence is very long and complicated, especially the clustering of adjectives like "intermediate-Mg calcite, benthic symbiont bearing, tropical foraminifer". I am not sure if it is correct to say "intermediate-Mg calcite foraminifer".

Line 100: This cannot be. Please double check the $H_2O_2$ concentration you used.

Line 120: Ass space after with.

Line 148: A clustering of terms like S40/27/2 within the main text is very confusing for the reader. I would suggest to reformat this part.

Line 157 to 163:

Of course the number of calcium counts would affect the element/Ca ratio if it would only vary alone. This would mean that this element is not bound in calcite since it is enriched in regions of lower calcite density. I am not sure if it is right to exclude these points from the dataset.

Line 170:

This is not an artificially grown inorganic crystal but biogenic precipitated calcite. Of course there are heterogeneities within foraminiferal test calcite.

Line 208: Delete space before Smirnov.

Line 246: I am not sure if it is correct to write "the groups produced specimens".

Line 247:

All not al.

Line 255: Add -4 to superscript. Also I would use another symbol than *.

Line 293: It should be possible to determine the genotype and especially regarding A. tepida I would strongly suggest to do this if you culture them.

Line 320: Again, I would use another symbol than *.

Line 329: Delete strong in brackets or the brackets.

Line 334: A complete reconstruction of past seawater conditions? Probably we would need a lot of more proxies for this.

Line 337: Two sentences in a row start with however.

Line 345: Several parts in this part of the discussion are repetitions from the last paragraph of the part before.

Line 348: Also here the aggressive cleaning procedure might have had different impact on different specimens. Line 352: There hasn't to be a linear correlation. If salinity exceeds a threshold of the "sweet spot" for a certain species this might influence the size outside this window...

Line 358: There is that new study by Fehrenbacher et al., 2017. They show that all chambers are uniformely overgrown by a thick layer of calcite even after the last chamber has been formed. If you do not find any significant trend between the last three chambers this might be worth to be discussed here.

Line 362: The precision gets better statistically, due to the higher amount of measurements but I agree if measurement time is limited and samples are abundant it would be

better to analyse more specimens rather than doing replicates on a single individual.

Line 388 and below: This part indeed is interesting but also quite speculative because nothing is known about ACC precursor absence or presence within these species as far as I know. I would tone this part a bit down making clear that the authors are hypothesizing.

Line 390: Write phase instead of phases.

Line 396: There are possibly more than these two mechanisms. Also the rest of the paragraph here is very speculative.

Line 406: Where is the primary organic sheet in figure 4? Aren't these bandings supposed to record day/night cycles during calcification (Fehrenbacher et al., 2017)

Line 409: It is not surprising that these bands are spatially correlated. Actually I think they are sitting in the same banding.

Line 410: Not Surprising?

Line 411: It should be possible to quantify this in your laser ablation profiles or don't you see these bands in there?

Line 403 and following: The discussion here is far too long and doesn't really come to a conclusive point.

Line 602: Figure 1: n=?

Line 625: Figure 4: POS is mentioned before it is defined in line 627 of the figure caption. Also it would be helpful if you mark in the picture what you think is the POS.

---

## Author Comment (AC1) · 15 Mar 2018

Dear Editor, Hereby we would like to thank the reviewers for their constructive comments, which helped improving the manuscript. Below we answer the questions and explain how we accommodated the changes suggested by the reviewers or in a few instances why we respectfully disagree. Reviewer comments are in italics, answers are normal font. Sincerely, also on behalf of the other co-authors, Esmee Geerken

Answers to anonymous Referee #2

General comments A paper by Geerken et al. reported variabilities in incorporation

ratios of Na and other elements to Ca under different salinity levels, observed in various scales of geochemi- cal anlalyses (intra-chamber, inter-chamber, inter-specimen, and inter-species levels). Overall, this paper is well-written, well-considered, and well-structured. The results would be useful for paleo-salinity reconstruction. However, I suggest that the authors consider the following criticisms and comments to improve the final version of this paper.

We thank the reviewer for the positive evaluation and appreciate the constructive criticisms and thoughtful comments. We elaborated upon the comments in more detail below and adjusted the manuscript accordingly.

The main criticism of this paper is a high inter-specimen variability of incorporation ratios despite using asexually reproduced juveniles incubated in the same petri dish. The authors explained this was caused by differences in the efficiency of calcification processes between specimens (L424-425). If so, what caused the efficiency of calcification process of the specimens? Physiological conditions of each specimen (nutrition, health, and food availability) and the microenvironment in culturing dishes may be clues to answer this question. These have not been measured in this study, but are partly reflected in growth rates of cultured specimens. I suggest the authors compare a relationship between TE/Ca and growth rates for each specimen, not comparing by their average values. This has been briefly described in L 243-248, but data and figures are not shown.

The high inter-specimen variability is a factor, which potentially complicates paleo-reconstructions, nevertheless such variability is observed for many foraminiferal culture studies (e.g. De Nooijer et al., 2014a). Apparently such inter-specimen variability is inherent to foraminiferal calcite. This variability could well be related to biomineralization, as the reviewer suggests, through differences in growth rates. Hence, we tested this by comparing shell sizes with individual shell chemistry. In this study the final shell size (assuming this represents growth rate during the experiment) is only for a few conditions statistically significantly correlated to El/Ca, and with a modest slope only. The fact that we did not find consistent results for these regressions made us decide not to add the figures, but merely mention this in the text. For reference we now added those figures to the appendix. Growth rates as presented in this figure, however, are not the same as calcite precipitation rates, hence how fast the calcite is precipitated during each chamber formation event rather then how many chambers are added during a given time period. The former might be biologically constrained and could indeed affect El/Ca ratios, based on what we know from inorganic precipitation experiments. We extensively address the topic inter-specimen variability in the discussion (section 4.2 and particularly lines 363-382) and also state that the actual cause of inter-specimen is currently not well understood and might be related to the observed heterogeneous El/Ca distribution within the shell.

Another criticism is a covariation between salinity and DIC. In this paper, these two parameters are positively correlated. The authors explained that this phenomenon is similar to the natural environment (L311-312), but this does not always occur in the natural environments (e.g. groundwater-seepage area). The authors could have manipulated the carbonate chemistry of culturing media to keep them constant and to make salinity the only variable. I suggest that the authors discuss the limitation of application for this proxy calibration to field specimens where salinity and DIC have not covaried/negatively covaried.

We agree that such experiments are needed to truly unravel salinity and sea water carbonate chemistry (alkalinity, DIC, etc). Previously, Wit et al. (2013) showed that with constant DIC Na/Ca still follows salinity, in experiments in which DIC was manipulated independently from salinity. Future experiments to further enhance the applicability and robustness of this new proxy, de-convolving alkalinity and salinity would be desirable, but go beyond the scope of the present manuscript. In the revised manuscript we will discuss limitations regarding potential co-variation of sea water carbonate chemistry and salinity for proxy calibration (lines 334 and further in the revised version of our manuscript).

The Discussion of Sections 4.2 and 4.3 should be combined and conclusions in this paper should be based on combined results of various analytical levels of all data sets including EPMA results. For example, Mg and Na are richly concentrated in organic layers based on EPMA results (Section 4.3). I wonder if this finding might affect your conclusions described in former sections.

We combined the sections accordingly, thereby integrating the different analytical approaches in the discussion. Intra-shell heterogeneity (with Mg and Na occurring at higher concentration close to or at the organic linings) is indeed likely an important factor for the observed inter-specimen differences. This is now mentioned at line 461-463 of the revised manuscript.

Specific comments L32-33: unclear what the authors mean to say. The sentence is removed and the previous sentence is slightly adjusted.

L35-39: need references References are added.

Section 2.1: Descriptions of two species are mixed up and make me confused. Better to separate if culturing conditions are different. For example, did two-third of all incubated specimens reproduced for both species? Were Ammonia specimens incubated at 25C in spite of their original cold habitat from the Wadden Sea? In addition, culturing protocol should be described after explaining culture media preparation (Section 2.2.1).

This has now been clarified in the revised manuscript at lines 114-117. In short, culture conditions were the same for both species and therefore we kept the culture method description combined. The order of the 'culture media preparation' and 'culture protocol' sections have been switched around as suggested.

L95-98: As far as my understanding, you changed culture media every week, then you put algal food and kept the culture media for one week. In that case, did the water quality (salinity and nutrients) change during a week? How did you seal the petri dishes to avoid the salinity changes?

A stock of freeze-dried Dunaliella was concentrated and diluted with seawater for each salinity treatment. Food was not entirely depleted after a week, therefore we assume that the growth of specimens was not hindered by a lack of food/nutrients. After a week specimens were transported to a new Petri dish to avoid accumulation of debris and food. Petri dishes were sealed with a lid, and water levels were monitored and found stable, so that salinity conditions remained similar within every week. We now clarified this in the text (lines 118-123).

L99-100: Does organic material removal affect the element ratio because Mg and Na are richly incorporated in POS? This question probably is based on a typographic error made in the original manuscript. We mentioned '70% H2O2' which should have been 1%. This was also the first comment of the second reviewer. The H2O2 step is intended to remove cell material from the shell and does not affect the element to calcium ratios (Barker et al., 2003). The fact that the structurally bound organics or trace elements are not affected by the oxidation step is supported by the clearly visible high Na/Ca and Mg/Ca bands within the test wall of cleaned specimens as revealed by EPMA (figures 4 of our manuscript). The lack of an effect of the cleaning procedure on the shell chemistry is now added to the Methods, lines 131-136.

L109: Describe the definition of salinity and how to measure it. Salinity was measured with a salinometer (VWR CO310), based on conductivity, and calibrated against standards (added in text, lines 90).

Table 1: Do these values indicate the mean values? Better to show the variations prior to and after changing culture media. These values are based on the stocks used to weekly renew the culture medium. Because we used the same identical stock solution throughout the experiment consistently of culture conditions could be guaranteed. Still, changes might have occurred through the week, but these are expected to have been minor. This is now mentioned at lines 125-130.

L127: Is a laser spot diameter small enough to measure the single chamber for small Ammonia specimens? We thank the reviewer for spotting this mistake. Ammonia specimens were actually measured with a smaller spot size of 60 $\mu$m to allow sampling individual chambers, this is now added to the manuscript (line 144).

Table 2: Explain Ac and Pr, and add the error. List the result of MACS-3 as well. I do not understand why accuracy is over 100%. We now define "Ac" and "Pr", which actually reflect the analytical error: Accuracy (Ac) and Precision (Pr). MACS-3 is the certified standard to determine accuracy, with values over 100% implying that the measured value is higher than the reference value. Often accuracy is expressed as a deviation from this desired 100% (the reference value).

L147-150: List them in Table or Appendix. Now listed in table 3

L243-248: Data and results are not presented. We added panels to figure 2, showing size and El/Ca relationships between groups. Individual El/Ca - size relationships within groups have been presented (regression results) in the text, line 272-279.

L246: need references Reference added to text.

L276: How did you identify POS from the SEM/EPMA maps? The organic linings are clearly observable on the backscatter SEM images (figure 4, appendix D). We assume the first organic lining must be the POS (now indicated on fig. 4 and Appendix D), but we changed 'POS' into 'organic linings' in the manuscript text because in A. lessonii, it is indeed not clear that the POS (which would be the first organic lining from the inside) coincides with the highest Mg/Ca peak.

L325-328: Do you have any ideas how inorganic carbon chemistry affect growth rates and shell weights? I think in future you have to more carefully check the physiology of cultured foraminifers to understand the proxy calibration, as described in the L364. We fully agree that this is an important unknown, yet actual precipitation rate is challenging to measure, especially on the scale of a larger culture experiment such as presented here. We do aim to look further into this on a smaller, individual foraminiferal scale in the future.

L329-334: I think this paragraph is meaningless, better to delete. We removed the paragraph.

L357-358: I think the latter sentence does not support the former sentence. We rephrased the paragraph.

Conclusions: This is just the summary of results. What did you think about these results? We adjusted the conclusions to better reflect the discussion.

Technical corrections

L56 (Allen et al., 2016): not italicized Adjusted

L117: delete : or ( Adjusted

L120: space between with and 0.14 Adjusted

L224: Fig. 3 should be Fig. 2 in the order of appearance. Adjusted

Figure3, L612: Results of A. lessonii should be placed on the left side for consistence throughout the paper. Adjusted

L613: Delete "for" after "regression" Adjusted

L234: average between inter-specimens? yes, the average of the average values of each specimen. Adjusted in text.

L235: unclear after = Adjusted

L243 and 247: replace al to all? Adjusted

L273: Fig. 6 must be Fig. 4. Adjusted

L274: Appendix D must be Appendix C. Adjusted

Table 3: What are A.l. and A.t.? A. lessoni (A.l.) and A. tepida (A.t), added to table description

Fig. 4: should be subdivided by a, b, c, d,. . . Adjusted

L625: transact > transect, No explanation for POS Adjusted

L627: POS should be indicated in the EPMA maps. Adjusted

L284-286: better to write "(for Ammonia tepida, Wit et al., 2013; for cultured Globigerinoides ruber, Allen et al., 2016; for field-collected G. ruber and G. sacculifer, Mezger et al., 2016)". Adjusted

L288-289: better to write ". . . reported previously for the planktonic G. ruber and G. sacculifer (e.g. Mezger et al., 2016; Allen et al., 2016)." Adjusted

L290: not targeted > non-target We think 'not targeted' is correct, we have however rephrased this sentence somewhat.

L329: Unclear expression: The absence of an (strong) impact of . . . We agree that this paragraph was unclear and have therefore removed it.

L329: Fig. 1 is related to this sentence? Paragraph has been removed.

L337: comma between 14% and 19% Adjusted

L347: RSD needs explanation (Relative Standard Deviation), added to text

L375: The difference in (an absence of) a correlation between elements between the two species > The difference or an absence of a correlation among elements between the two species? We rephrased the paragraph.

Answers to Anonymous Referee #3

The study by Geerken et al. reports the influence of salinity on Na/Ca, Mg/Ca and Sr/Ca of the benthic foraminiferal species A. lessonii and A. tepida, with a focus on Na/Ca as a possible salinity proxy. Element ratios have been determined by laser ablation ICP-MS and foraminiferal specimens have been collected from cultures with different different salinities. The authors found a significant correlation between salinity and Na/Ca and additionally made elemental mappings of the different elements on cross-sections of selected specimens with electron microprobe.

First of all I have to say that I was impressed to see the amount of data and work which has been put into the analyses and culturing. More than 200 specimens have been analysed with more than 600 measurements! I regret to say after reading the manuscript carefully that there is a substantial problem with this study and at least major revisions are required for the manuscript. The problem is already mentioned by the authors themselves in the first paragraph of the discussion: The cleaning procedures. The authors mention differences to former calibrations of Na/Ca as salinity proxy within the same species (A. tepida). Wit et al. (2013) used 5% sodium hypochloride to bleach the forams and remove organic contaminations. Geerken et al. now report that they bleached the forams in 70% $H_2O_2$. I do not think this can be true. Most of the $H_2O_2$ solutions which are commercially available are only 30%. In this case it would be 70% of a 30% solution which would be around 21%. This is still much to high! There are a lot of studies about cleaning procedures for analyses on foraminifera and standard methods have been developed based on studies which started already in the 1980s. The standard method for organic matter removal, which is widely applied in most of the labs, is bleaching in a 1% $H_2O_2$ solution (Barker et al., 2003). Barker et al.(2003) also showed that there is a significant impact of different oxidative treatments on foraminiferal Mg/Ca, though not on Sr/Ca ratios. Why did the authors not apply these standard cleaning techniques? The SEM pictures within appendix C already show that, especially A. tepida appears to be not very well preserved anymore after the treatment. The EMP mappings shown in figure 4 also indicate chemical alteration on the outer part of the test walls of A lessonii (see Mg/Ca mapping and SEM picture). While most of the standard cleaning procedures are used for analyses of bulk samples with ICP-MS, I am not aware if there is any study on the impact of different cleaning procedures on laser ablation analyses. Are there studies at all on different cleaning procedures regarding microanalyses? Otherwise I would strongly recommend to develop a standard treatment also for laser ablation studies. Only this would ensure comparability between different studies. From this point of view I think the authors cannot provide a quantitative calibration for Na/Ca ratios as a salinity proxy. They can show that there is a significant relative effect of salinity on Na/Ca and thus give further evidence for the validation of this proxy. I suggest to rewrite the manuscript bearing this in mind.

We thank the reviewer for the recognition of the large amount of analytical work condensed into this manuscript. We sincerely apologize for a typographic error, on which a major comment of the author hinges. "70% H2O2" should have been '1%', since in our group we always use the same recipe (e.g. Mezger et al., 2016; Van Dijk et al., 2017, Biogeosciences; Van Dijk et al., 2017, EPSL), based on an adapted oxidation step from the Barker et al.'s (2003) cleaning protocol (we replace the buffer NaOH with NH4OH to avoid Na-contamination of our specimens since we study Na/Ca). We understand that the referee was concerned about this (impossibly high) H2O2 and potential impact on the quality of our cleaned samples. We have corrected this in the revised version of our manuscript (line 134).

General: There are a lot of repetitions in the text, especially in the discussion. The manuscript is quite long and would benefit a lot by shortening some parts of the text. We fully agree that a manuscript/paper should be as condensed as possible and already tried to structure the text accordingly. We went through the text again carefully to omit potential repetitions. We rephrased and condensed where possibly: marked in yellow in the revised version of our manuscript.

Line 46 - 50: Nürnberg et al., (1996) was on Mg/Ca and not on Sr/Ca as far as I know. The reference has been removed.

Page 71 - 74: Please rewrite. This sentence is very long and complicated, especially the clustering of adjectives like "intermediate-Mg calcite, benthic symbiont bearing, tropical foraminifer". I am not sure if it is correct to say "intermediate-Mg calcite foraminifer". The sentence has been split into two separate sentences (line 73-77). Intermediate-Mg calcite is a commonly used reference to species making calcite with Mg/Ca between 5-10% (Bentov and Erez, 2006).

Line 100: This cannot be. Please double check the H2O2 concentration you used. We apologize again for this confusion, and thank the reviewer for noticing. Concentration of H2O2 has now been adjusted to the used 1% value (line 134).

Line 120: Ass space after with. Adjusted in text

Line 148: A clustering of terms like S40/27/2 within the main text is very confusing for the reader. I would suggest to reformat this part. This information is now added to table 3.

Line 157 to 163: Of course the number of calcium counts would affect the element/Ca ratio if it would only vary alone. This would mean that this element is not bound in calcite since it is enriched in regions of lower calcite density. I am not sure if it is right to exclude these points from the dataset. When Mg substitutes for Ca, by definition this lowers the Ca content of the ablated material (although to a very limited extend), but it is still calcite bound. Still, this could potentially result in a closed sum effect, thereby causing artificial co-variation of Na/Ca and Sr/Ca, see also discussion in Van der Weijden (2002). This is especially of concern for species with higher Mg/Ca contents, such as A. lessonii, which is addressed at lines 187-196. Other Ca-variation (e.g. due to lower Ca-density) could also result in spurious correlations between element to calcium ratios and, therefore, we performed a Monte Carlo simulation, testing whether the coupled inter-specimen trends in Na/Ca and Sr/Ca versus Mg/Ca could be the result of such a spurious correlation. We now tried to clarify this in the text of our revised manuscript (lines 183-186).

Line 170: This is not an artificially grown inorganic crystal but biogenic precipitated calcite. Of course there are heterogeneities within foraminiferal test calcite. We agree

with the reviewer on this point, but feel we have to rule out any potential analytical bias. Especially since we address inter-elemental trends and link these to biomineralization mechanisms in the discussion, we need to rule out spurious correlations and closed-sum effects.

Line 208: Delete space before Smirnov. Adjusted accordingly.

Line 246: I am not sure if it is correct to write "the groups produced specimens". The sentence has been rephrased.

Line 247: All not al. Adjusted.

Line 255: Add -4 to superscript. Also I would use another symbol than *. Adjusted. We agree and changed '*' into '$\times$' throughout the document.

Line 293: It should be possible to determine the genotype and especially regarding A. tepida I would strongly suggest to do this if you culture them. This has been tested for A. tepida indeed, and the genotype of A. tepida we use in experiments is T6 (de Nooijer et al., 2014). We now added this information to the method section of the new manuscript (line 109)

Line 320: Again, I would use another symbol than *. Adjusted throughout the manuscript into '$\times$'.

Line 329: Delete strong in brackets or the brackets. Deleted "(strong)".

Line 334: A complete reconstruction of past seawater conditions? Probably we would need a lot of more proxies for this. Changed "complete" into "a suite of"

Line 337: Two sentences in a row start with however. Changed first "however" in "nevertheless"

Line 345: Several parts in this part of the discussion are repetitions from the last paragraph of the part before. We have rewritten this part of the discussion, now avoiding unnecessary repetition.

Line 348: Also here the aggressive cleaning procedure might have had different impact on different specimens. The '70%' was a typo, as indicated before.

Line 352: There hasnot to be a linear correlation. If salinity exceeds a threshold of the "sweet spot" for a certain species this might influence the size outside this window... We agree that a linear relation is actually rather unlikely. There rather seems to be a growth optimum for salinity, which has now been added to the text. A linear correlation between salinity and size was suggested by Wit et al. (2013), therefore we still address a linear regression test. We have a figure (Fig. 2) in the manuscript with shell sizes per salinity condition, to give the reader an appreciation of foraminiferal growth during the experiment.

Line 358: There is that new study by Fehrenbacher et al., 2017. They show that all chambers are uniformly overgrown by a thick layer of calcite even after the last chamber has been formed. If you do not find any significant trend between the last three chambers this might be worth to be discussed here. Although an interesting concept, the benthic species from our study calcify morphologically different from the planktonic foraminiferal species N. dutertrei used in Fehrenbacher et al. (2017). A. lessonii and A. tepida precipitate their calcite in a laminar growth pattern, without the formation of a terminal, thick layer encapsulating the complete specimen. This is also different from calcification in Orbulina universa, where a gradual, thick spherical chamber is formed, characterized by alternating high- and low-Mg element bands.

Line 362: The precision gets better statistically, due to the higher amount of measurements but I agree if measurement time is limited and samples are abundant it would be better to analyse more specimens rather than doing replicates on a single individual. Accuracy is improved by increasing number of measurements on single specimens, but not the precision of the salinity estimate. Importantly, analyzing a sufficiently high number of specimens is (statistically) crucial, considering the high inter-specimen variability due to vital effects. We tried to clarify this in the text (lines 381-388).

Line 388 and below: This part indeed is interesting but also quite speculative because nothing is known about ACC precursor absence or presence within these species as far as I know. I would tone this part a bit down making clear that the authors are hypothesizing. We acknowledge that this part is rather speculative, although the recent observation of a vaterite precursor step makes an ACC precursor more likely (Jacob et al., 2017). We now emphasize that these ideas are strictly hypothetical and also clarify that some observations are based on inorganic precipitation experiments.

Line 390: Write phase instead of phases. Adjusted

Line 396: There are possibly more than these two mechanisms. Also the rest of the paragraph here is very speculative. This has been adjusted to: 'these (and other) potential mechanisms' and we down toned the paragraph by adding the last sentence: "However, since there are many, both biotic and abiotic, mechanisms that can (simultaneously) influence (coupled) element partitioning, it is difficult to resolve the exact mechanism behind inter-specimen and inter-species differences in El/Ca".

Line 406: Where is the primary organic sheet in figure 4? Aren't these bandings supposed to record day/night cycles during calcification (Fehrenbacher et al., 2017) We now indicate the location of the POS in fig 4. As mentioned before, calcification in Ammonia and Amphistegina deviates slightly from that in N. dutertrei. Nevertheless, chamber addition and hence element banding in the latter species could indeed be diurnally paced, which remains to be investigated. We added these remarks to the text (lines 448-451).

Line 409: It is not surprising that these bands are spatially correlated. Actually I think they are sitting in the same banding. We are not sure whether we fully understand this remark, but with 'spatially correlated' we do refer to 'sitting in the same band'.

Line 410: Not Surprising? Changed to 'makes sense' and 'lower' for A. tepida. It is not surprising because if overall Mg/Ca in A. tepida is very low you would not expect pronounced Mg banding (simply because there is no Mg).

Line 411: It should be possible to quantify this in your laser ablation profiles or don't you see these bands in there? We do see variability in our laser ablation profiles, however, the resolution is much lower less then the profiles based on the EPMA maps, due to mixing of the signal. A depth-profiling La-ICP-MS technique has indeed been developed by Eggins and Sadekov, and we aim to apply this approach in the future (or use a standard to quantify EPMA maps). At this moment the specimen are embedded in resin so it is not possible anymore to do quantify bands.

Line 403 and following: The discussion here is far too long and doesn't really come to a conclusive point. We tried to be more conclusive and concise in the revised version.

Line 602: Figure 1: n=? Line 625: Adjusted in text.

Figure 4: POS is mentioned before it is defined in line 627 of the figure caption. Also it would be helpful if you mark in the picture what you think is the POS. Answer: we changed this accordingly, and added the position of the POS in the SEM image.

Please also note the supplement to this comment:
https://www.biogeosciences-discuss.net/bg-2017-481/bg-2017-481-AC1-supplement.pdf

[Figure]

**Supplement:**

[revised manuscript text omitted]